# Spatial and Temporal Pattern of Norovirus Dispersal in an Oyster Growing Region in the Northeast Pacific

**DOI:** 10.3390/v14040762

**Published:** 2022-04-06

**Authors:** Timothy J. Green, Chen Yin Walker, Sarah Leduc, Trevor Michalchuk, Joe McAllister, Myron Roth, Jasmine K. Janes, Erik T. Krogh

**Affiliations:** 1Faculty of Science and Technology, Vancouver Island University, Nanaimo, BC V9R 5S5, Canada; chenyin.walker@viu.ca (C.Y.W.); sarah.leduc@viu.ca (S.L.); trevor.michalchuk@viu.ca (T.M.); joemcallister3@gmail.com (J.M.); jasmine.janes@viu.ca (J.K.J.); erik.krogh@viu.ca (E.T.K.); 2BC Ministry of Agriculture, Food & Fisheries, P.O. Box 9120, Victoria, BC V8W 9B4, Canada; myron.roth@gov.bc.ca; 3Department of Ecosystem Science and Management, University of Northern British Columbia, Prince George, BC V2N 4Z9, Canada

**Keywords:** oyster, norovirus, environmental transmission, non-point source, coastal waters

## Abstract

Contamination of Pacific oysters, *Crassostrea gigas*, by human norovirus (HuNoV) is a major constraint to sustainable shellfish farming in coastal waters of the Northeast Pacific. HuNoV is not a marine virus and must originate from a human source. A barrier to effective management is a paucity of data regarding HuNoV dispersal in the marine environment. The main objective of this study was to identify the spatial distribution and persistence of HuNoV in an active shellfish farming region in the Northeast Pacific. Market-size *C. gigas* were sequentially deployed for two-week intervals at 12 sites during the 2020 winter risk period from January to April. Detection of HuNoV quantification was performed by reverse transcription real-time PCR (RTqPCR) according to method ISO 15216-1:2017, with modifications. RTqPCR did not detect GI HuNoV. The estimated prevalence of GII HuNoV in oyster digestive tissue was 0.8 ± 0.2%. Spatiotemporal analysis revealed that contamination of oysters with GII HuNoV changed through time and space during the surveillance period. A single cluster of oysters contaminated with GII.2 HuNoV was detected in a small craft harbor on 23 April. There was no significant increase in the proportion of positive pools in the next nearest sampling station, indicating that HuNoV is likely to disperse less than 7 km from this non-point source of contamination. Results from this study indicate that HuNoV contamination of coastal waters from non-point sources, such as small craft harbors and urban settings, can pose a significant localised risk to shellfish farming operations in the region.

## 1. Introduction

Norovirus is a significant public health problem in North America [1]. It is estimated that ~1 million Canadians experience foodborne gastroenteritis linked to norovirus each year [2]. Noroviruses are a group of genetically diverse viruses belonging to the genus *Norovirus*, family *Caliciviridae* [3,4]. There are at least 49 norovirus genotypes classified within ten genogroups (GI to GX) [3]. Genogroups GI and GII are more frequently associated with human illness [3]. Human noroviruses (HuNoV) are highly infectious, with just 10 to 100 virions required to induce disease [5], and transmission primarily occurs via person-to-person contact, aerosolized particles from vomites, or ingesting contaminated food or water [6].

There have recently been several notable outbreaks of HuNoV across Canada and the USA attributed to the consumption of oysters harvested from Western Canada [7,8]. HuNoV is not a natural marine virus and must originate in coastal waters via human faeces or vomit [9]. Oysters are filter-feeders and can bioaccumulate enteric viruses in their tissues with a concentration factor of >99-fold compared to surrounding seawater [10]. Noroviruses persist in oyster tissue for long periods by binding to specific antigens in the gill and digestive gland tissue [11]. Oysters are often consumed raw or lightly cooked, posing a significant risk if oysters are cultivated in areas contaminated by human sewage [12]. Sources include effluent from wastewater treatment plants, combined sewer/stormwater overflows, malfunctioning septic tanks, and recreational and commercial fishing vessels [9]. Public health officials estimate that between 10,900 and 552,000 HuNoV illnesses are linked to consumption of raw oysters occur every year in Canada [12], but these values are difficult to estimate using existing epidemiological data, and thus characterized by a large amount of uncertainty. Although the focus of this study is on Canada, HuNoV contamination of oyster is a global problem with the United Kingdom reporting ~11,800 cases of HuNoV per year linked to consumption of contaminated oysters [13].

The Canadian shellfish sanitation program (CSSP) is administered by the Canadian Food Inspection Agency (CFIA), Department of Fisheries and Oceans Canada (DFO), and Environment and Climate Change Canada (ECCC) to classify and monitor shellfish harvest areas and ensure shellfish are safe for human consumption. The CSSP manual [14] is a reference document for monitoring, classifying and controlling areas where bivalve molluscan shellfish are harvested. The CSSP manual defines various exclusion zones, where shellfish may not be harvested around point sources, such as wastewater treatment outfalls, small craft harbors, and floating accommodation. The CSSP also undertakes surveys, including shoreline sanitary investigations and bacteriological monitoring of water and shellfish, to classify the suitability of areas for shellfish harvesting. The CSSP has various regulatory tools to close areas to shellfish harvesting during significant weather events, sanitary wastewater discharges and traceback of reported human illness linked to shellfish consumption. The criteria to reopen harvest areas include a minimum period of time to allow shellfish to purge human pathogens and meeting requirements of allowable fecal coliform and HuNoV levels in water and shellfish samples. If CFIA detects norovirus in any single oyster sample, the shellfish harvest area will be closed for a minimum of 30 days

In the recent outbreaks of norovirus linked to raw oyster consumption in western Canada, no environmental pollution source(s) could be identified as the cause of the outbreak [7,8]. Given the low infective dose and the viability of HuNoV in cold waters, it has been postulated that wastewater effluent spread by ocean currents can contaminate geographically dispersed oyster farms in western Canada [12,15]. Field data to substantiate widespread dispersal of HuNoV in the marine environment of the Northeast Pacific are lacking, and this surveillance study was undertaken to provide data for the shellfish industry, and its regulators, on the temporal and spatial distribution of HuNoV in an oyster farming region in western Canada.

## 2. Materials and Methods

### 2.1. Field Study

This study was conducted in a coastal Sound that produces ~5600 tonnes of Pacific oysters with a landed value of CAD$ 10.31 million [16]. The Sound is approximately 25 km long and is 3.5 km wide at its widest point, with the average width being less than 2 km (Bendell and Wan, 2011). The surface area of the Sound is 87 km^2^ and the residence time for bottom water is ~2 months with the majority of tidal exchange occurring at the southern end of the Sound [17]. Anthropogenic disturbances include two foreshore cities at the northern end of the Sound and a number of rural villages on septic that fringe the Sound (Figure 1). Wastewater from ~44,000 residents of the northern cities is processed by a wastewater treatment plant and effluent is discharged into a body of water outside the Sound.

All sentinel oysters used in this study were from a single batch of two-year-old *Crassostrea gigas* produced by the shellfish hatchery at Vancouver Island University. Prior to the start of the trial, oysters were divided between triplicate 4000 L depuration tanks. No seawater exchange occurred in depuration tanks to ensure oysters were free of HuNoV and fecal coliforms. A subset of 45 oysters were confirmed to be free of GI and GII HuNoV prior to commencement of the trial according to methods outlined below. A total of 2520 oysters were deployed over a 12-week period between January 2020 and April 2020 at 12 sampling-sites over 6 sampling periods. Sampling sites consisted of active shellfish farms and prohibited harvest areas, such as small craft harbors (i.e., marinas) and recreational beaches (Figure 2). Oysters were divided into lots of 35 oysters, placed in pearl nets and sequentially deployed at two-week intervals. At the start of the trial, a pearl net containing 35 oysters was deployed at each sampling-site. After two weeks, the pearl nets containing the oysters were retrieved and a second batch of 35 oysters were deployed at each site. This pattern of two-week sequential deployment of oysters at each site was repeated until the end of the trial. Retrieved oysters were frozen at −80 °C. Temperature, salinity, pH and dissolved oxygen concentration at each site was recorded using a Pro Plus multi-parameter water quality instrument (YSI, Yellow Springs, OH, USA).

### 2.2. Norovirus Analysis

Oysters were tested for GI and GII HuNoV according to the ISO 15216-1_2017-03 method “microbiology of food and animal feed—horizontal method for determination of HAV and NoV in food using real-time PCR” with the exception that bacteriophage MS2 (ATCC^®^ 15597-B1^TM^) was used as the process control virus. Oysters were thawed at 4 °C overnight prior to sample processing. Digestive tissue (DT) from 5 oysters was pooled (7 pools per site) and placed in a 30 mL tube with screw cap (OMNI #19-6635), equal volume of proteinase K solution (30 units/mL; Invitrogen^TM^ #AM2542) added to each sample before homogenizing using 3 × 5 mm stainless steel beads (Qiagen #69989) and Fisherbrand^TM^ bead mill 24. Dissection tools were disinfected between samples using 5% bleach solution for 5 min followed by rinsing in municipal tap water. Following incubation in a New Brunswick E-24R orbital shaker at 300 rpm and 37 °C for 45 min, samples were centrifuged at 1500× *g* for 5 min. The supernatant was retained at −80 °C for nucleic acid extraction.

Purification of viral RNA was carried out using the MagMax^TM^ 96 total RNA Isolation Kit (ABI #AM1830) following the manufacturer’s recommendation. In brief, viral RNA was extracted using guanidine isothiocyanate, absorbed onto magnetic silica beads, washed with various buffers, DNA eliminated with DNase I, and RNA released into 30 µL of elution buffer. Each batch of RNA extractions included a negative control (sterile water) and each sample was spiked with MS2 process control. First-strand synthesis was performed with iScript^TM^ Select cDNA synthesis kit (Bio-RAD #1708897) using random hexamers. Primer and probes for HuNoV GI, HuNoV GII and MS2 process control virus are listed in Table 1. Real-time PCR was performed in a CFX384 Real-Time Touch Thermocycler (Bio-Rad, Hercules, CA, USA) with master mix and template dispensed by an epMotion^®^ 5073 m liquid handling work station (Eppendorf, Hamburg, Germany). Thermocycling parameters were 3 min at 95 °C followed by 40 cycles of 10 s at 95 °C, 30 s at 60 °C. qPCR was performed in duplicate for both samples and standards. Ten-fold serial dilutions (10^5–^10^1^ copies) of plasmid containing the target region were included on each qPCR plate as a standard curve. A valid run was defined as a run exhibiting no amplification of the negative control and standard curve with r^2^ > 0.95 and efficiency between 85 and 110%. The dynamic range of the standard curves was 5 × 10^5^ to 5 × 10^1^ copies. Samples were considered valid when the extraction efficiency of MS2 was greater than 5% and the amplification efficiency of MS2 was >85%, calculated from the slope of the curve between neat and five-fold diluted cDNA sample. Samples that did not meet the extraction- or amplification-efficiency for MS2 were re-extracted and tested again from oyster digestive tissue stored at −80 °C. Re-extracted samples that did not meet the criteria for extraction- or amplification-efficiency were not considered for statistical analysis. Samples were defined as positive when both technical replicates exhibited an exponential accumulation of fluorescence (Cq value < 40) and when a sample exhibited one replicate positive and one replicate negative, it was considered to be “inconclusive”.

### 2.3. Norovirus Confirmation

Confirmation of GII HuNoV in RTqPCR positive samples was conducted using a combination of RT-nested PCR, cloning and nucleotide sequencing. Primers COG2F and G2SKR were used for the first round of RT-PCR. G2SKF and G2SKR primers were used for the nested round of RT-PCR with high fidelity iProof polymerase master mix (BioRad #1725310). PCR products were purified using QIAquick PCR Purification Kit (Qiagen #28104), ligated into the pCR^TM^ 2.1-TOPO^TM^ vector, and transformed into One Shot TOP10 chemically competent *E. coli* (Thermo Fisher #K450002). Positive clones were forward and reverse sequenced at the Molecular Biology Facility, University of Alberta using BigDye terminator 3.1 (Applied Biosystems, Waltham, MA, USA). Nucleotide sequences were checked for sequencing errors and vector contamination removed using Sequencher v5.4.1 (Gene Codes Corporation, Ann Arbor, MI, USA). Genotypes were assigned with the NoV Noronet typing tool (http://www.rivm.nl/mpf/norovirus/typingtool accessed on 28 May 2021) using default parameters [18].

### 2.4. Escherichia coli Analysis

Genomic DNA was purified from pooled digested DT samples using the DNeasy Blood Tissue Kit (Qiagen #69506) and eluted in 100 µL of sterile water. Contamination of oysters with *E. coli* was determined by qPCR targeting the ybbW gene (Walker et al., 2017). The ybbW gene is part of the *E. coli* ‘core genome’ and consists in >95% of all sequenced *E. coli* (Walker et al., 2017). Primer sequences are listed in Table 1. Real-time PCR was performed in a CFX384 Real-Time Touch Thermocycler (Bio-Rad) using a total reaction volume of 15 µL containing SsoFast^TM^ EvaGreen^®^ Supermix (Bio-RAD #1725204), primers listed in Table 1, and 5 µL of template. qPCR was performed in duplicate and a Ct < 40 was considered a positive sample. Ten-fold serial dilutions (10^5–^10^1^ copies) of plasmid containing the target region were included on each qPCR plate as a standard curve.

### 2.5. Pathogen Prevalence and Correlation to Environmental Variables

The prevalence (±standard error) of oysters contaminated with GII HuNoV or *E. coli* at each site was estimated from the number of positive pools following a Bayesian approach [19] employing a pooled prevalence calculator with default parameters (http://epitools.ausvet.com.au/ppfreqthree, accessed on 20 May 2021) for fixed pool size and tests with uncertain sensitivity and specificity. The correlation between prevalence of HuNoV and *E. coli* in oyster DT to environmental variables (sea surface temperature, salinity, pH and dissolved oxygen) was tested using Pearson’s correlation in Graphpad Prism version 9.3.1.

### 2.6. Spatial and Temporal Analysis

Retrospective analysis for clustering of GII HuNoV in both space and time was performed using the Bernoulli model available in SaTScan v9.6.1 [20]. A positive case was a pool positive for GII HuNoV and a control case was a pool negative for GII HuNoV. The model detects spatiotemporal clusters using a cylindrical window that moves across a map of specified longitude/latitude coordinates, where the width of the cylinder represents space and the height reflects time. The size of the cylindrical window can fluctuate during the scan and was set in this study to a maximum spatial cluster size of 50% and temporal cluster size of 90% (maximum available option). This allowed clusters to remain present over most of the study period. To determine statistically significant clusters, a likelihood ratio test was performed by comparing the number of observed cases in a cluster to the number of expected cases if they were randomly distributed through space and time. Monte Carlo hypothesis testing (999 simulations) was used to obtain *p*-values given a significance level of alpha < 0.05.

## 3. Results

### 3.1. Description of Field Data

From 22 January to 23 April 2020, a total of 2415 oysters (in lots of 35) from twelve sites were assessed by qPCR for HuNoV and *E. coli* in pools of five oysters (site 12 was missed on three occasions due to tidal access constraining boat sampling). *E. coli* was detected in 1.3% of pooled oyster DT samples by qPCR. GI HuNoV was detected in 0% of samples by RTqPCR. GII HuNoV was detected in 4.0% of pooled samples and detection of GII HuNoV was inconclusive in 13.5% of samples by RTqPCR. The Cq values for positive GII HuNoV samples ranged between 33.7 and 39.8 (average 37.3). The estimated prevalence (±standard error) of GII HuNoV in oyster DT over the entire surveillance period was 0.8 ± 0.2%. There was no difference in the estimated prevalence for GII HuNoV at active shellfish farm sites (0.8 ± 0.3%) compared to recreational beaches (0.6 ± 0.3%) and small craft harbors (1.0 ± 0.4%). The estimated prevalence for GII HuNoV in the Sound changed throughout time, with the fifth sample-period (0.1 ± 0.8%) having the lowest prevalence and the sixth sample period (6.8 ± 1.7%) having the highest prevalence. Figure 1 shows the estimated prevalence for GII HuNoV in the Sound using both positive and inconclusive cases. No correlation was observed between the prevalence of GII HuNoV in oyster DT and seawater temperature, salinity, pH, oxygen concentration or prevalence of *E. coli* (*p* > 0.05).

### 3.2. Cluster Analysis

The spatiotemporal analysis detected one significant cluster of GII HuNoV in oyster DT within the study area between 8 April and 23 April 2020 (Table 2). Figure 2 shows the geographical distribution of the cluster, which occurred at a single sample station (NV1). The next nearest sample station (NV2), which is a 7.1 km away, did not have a significant increase in the frequency of GII HuNoV positive pools. This significant cluster occurred in a small craft harbour adjacent to an urbanised area. The temporal window of the cluster was ≤2 weeks, but surveillance in the Sound ended at this time-point. Nested RT-PCR followed by DNA sequencing confirmed the presence of GII.2 HuNoV.

The spatiotemporal analysis was also repeated with the inconclusive samples as positive cases in the model. Again, only one significant cluster of GII HuNoV was identified at NV1 between 8 April and 23 April 2020 (*p* = 0.032).

## 4. Discussion

This study aimed to identify the temporal and spatial distribution of HuNoV in a coastal Sound in the Northeast Pacific. Surveillance was undertaken during winter, when risk of illness from raw oyster consumption is highest [9,21]. The study was designed to identify HuNoV dispersal from non-point sources by sequentially deploying depurated oysters at bi-weekly intervals at 12 locations within the Sound. Our results indicate that the prevalence of HuNoV in Pacific oysters was low (0.8 ± 0.3%) during the surveillance period and we never detected GI HuNoV in oyster digestive tissue. The spatiotemporal analysis identified a single non-point source contamination event in a small craft harbor that is adjacent to an urban center. Dispersal of HuNoV was likely to be less than 7 km as the next nearest sample station did not observe an increase in the frequency of HuNoV (Figure 2). These observations are highly relevant to the shellfish industry and its regulators as it highlights the risk of non-point sources in the Sound. Local government and the shellfish industry can also educate users of the small craft harbor on appropriate disposal of human sewage.

Modelling indicates that prevalence, rather than levels, of HuNoV in oysters drives the risk of human illness from consumption of raw oysters [12]. We used a pooled sample approach to estimate prevalence of HuNoV in oysters. Testing pooled samples is routinely undertaken in surveillance programs when aquatic animals do not display clinical signs and prevalence in the population is low, and to reduce costs with testing a larger proportion of the population [22]. During the period of time that we conducted our field survey, estimated prevalence of HuNoV in oysters was estimated to be 0.8 ± 0.3%. For perspective, the prevalence of HuNoV in oysters from other regions is reported in the range of <2% in Australia [23], 9% in France [24], 3.9 to 20% in the USA [25,26], 16.9% in China [27], 32.1% in Spain [28], and up to 71.6% in the United Kingdom [29].

The low prevalence of HuNoV in our study is supported by the amount of GII HuNoV RNA in positive samples, which was often below the limit of quantification of the RTqPCR assay (average Cq value = 37.3). The presence of GII HuNoV could only be confirmed at one site by nested RT-PCR and DNA sequencing (Table 2). It is important to take into account that 13.5% of pooled samples were inconclusive with one replicate positive and one replicate negative for GII HuNoV by RTqPCR. Large variability in technical replicates can occur in RTqPCR when target cDNA is very dilute [30]. Thus, it is plausible that we underestimated the true prevalence of HuNoV in the Sound. Furthermore, RTqPCR detects the viral genome of HuNoV, and is unable to discriminate between infectious and non-infectious viral particles [31]. Thus, it is not currently possible to determine the human health risks posed by consuming raw oysters contaminated with different levels of HuNoV; consequently the Canadian shellfish sanitation manual [14] recommends a shellfish harvest area be closed for a minimum of 30 days if any single oyster sample is positive for HuNoV. In the United Kingdom, the amount of HuNoV in outbreak-related oyster samples (i.e., strongly linked to HuNoV illness) is typically a magnitude higher than the amount of HuNoV found in non-outbreak-related oyster samples (i.e., oysters collected from a commercial shellfish farms) [29]. This observation helps to explain the disparity between the high proportion of farmed oysters contaminated with HuNoV in the United Kingdom and the relatively low number of epidemiologically confirmed outbreaks of HuNoV linked to raw oyster consumption [21,29].

Our spatiotemporal analysis detected one significant GII HuNoV cluster during the study period. The finding that GII HuNoV clusters in space and time suggests a higher risk for human consumption of oysters in these areas compared to other locations during the study. The significant cluster was confirmed to be GII.2 HuNoV by nested RT-PCR and DNA sequencing, and occurred in a small craft harbor (site: NV1) that is potentially contaminated from illegal vessel discharges or run-off from the local urban environment. NV1 is located within a prohibited area for shellfish harvesting as it is within 125 m of a marina [14]. NV1 is separated from the nearest sample site (NV2) by a distance of 7.1 km, indicating that the extent of the impacted area was likely to be less than this distance. Our surveillance of the Sound ended at this sampling-point, preventing us from determining how long HuNoV persists in the water column. The spatiotemporal clustering in our study corroborates findings from other studies investigating the zone and duration of HuNoV dispersal in the marine environment from known point-sources. In Australia, GII HuNoV was detected in Sydney rock oysters up to 5.29 km downstream and persisted in oysters for 6 weeks following a pump station sewage overflow event [32]. In another study conducted in the USA, GII HuNoV could be detected as far as 5.74 km from the discharge location of a wastewater treatment plant [33].

One potential limitation of this study was that surveillance in the Sound overlapped with the start of the 2020 global pandemic of SARS-CoV-2. The incidence of HuNoV illness in North America significantly declined in April 2020 due to nonpharmaceutical interventions of physical distancing, mask wearing, surface disinfection, and increased hand hygiene to control the SARS-CoV-2 global pandemic [34]. The impact of these public health measures on HuNoV prevalence in the population surrounding the Sound is difficult to quantify. We did detect GII.2 HuNoV within the Sound on 23 April 2020 and this sample period coincided with the highest prevalence of GII HuNoV in the Sound (Figure 1). During the first wave of SARS-CoV-2 (May 2020), French oysters from the Atlantic and Mediterrean coasts were also contaminated with HuNoV [35].

## 5. Conclusions

In summary, we observed a low prevalence of HuNoV in oysters within a Sound in western Canada, but prevalence changed through time and space, indicating that non-point sources, such as marinas and urban runoff, were potential sources of contamination of farmed shellfish. Analysis indicates that the area and duration of HuNoV contamination in the marine environment is likely to be <7 km. These data should support science-based decisions on shellfish closures following outbreaks of norovirus illness associated with raw oyster consumption. Further studies incorporating hydrodynamic models to inform sampling locations within the Sound may allow predictions of norovirus transmission in the marine environment to be improved.

## Figures and Tables

**Figure 1 viruses-14-00762-f001:**
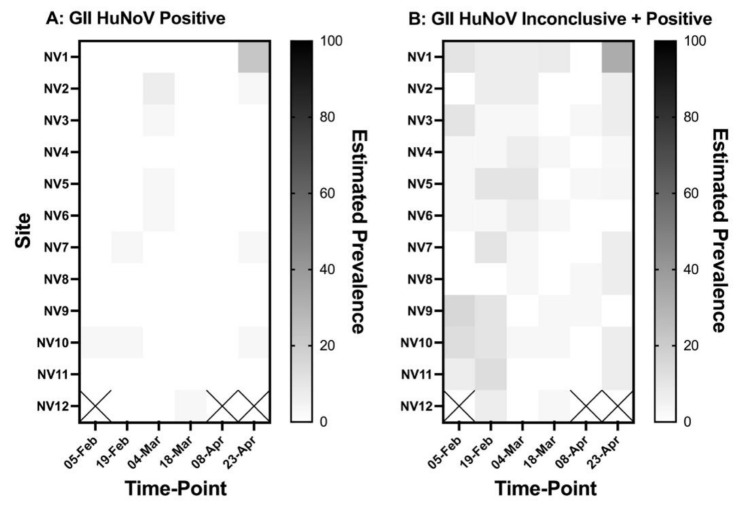
Change in the estimated prevalence for GII HuNoV between sample locations and time-points. Prevalence was calculated for both positive cases (**A**) and positive and inconclusive cases combined (**B**). The locations where sentinel oysters were deployed is presented in Figure 2.

**Figure 2 viruses-14-00762-f002:**
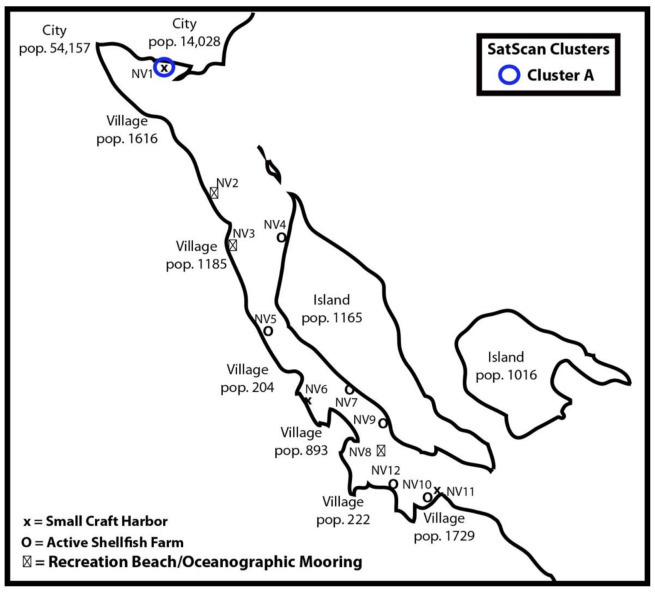
Schematic representation of sample sites within the Sound. One significant (cluster A: blue circle) was identified in the Sound using SatScan v9.6.1. Sentinel oyster deployment locations are marked as NV1–NV12. Locations marked as “o” are active shellfish farms, “x” are small-craft harbors, and ☒ are areas closed to shellfish harvesting.

**Table 1 viruses-14-00762-t001:** Primers and probes sequences, and source for RTqPCR assays used in this study. Probes were labelled with 5′-carboxyfluorescein (FAM) and 3′-black hole quencher-1 (BHQ-1).

Target	Forward	Probe	Reverse	Reference
HuNoV GI	CGCTGGATGCGNTTCCAT	TGGACAGGAGATCGC	CCTTAGACGCCATCATCATTTAC	ISO 15216-1
HuNoV GII	ATGTTCAGRTGGATGAGRTTCTCWGA	AGCACGTGGGAGGGCGATCG		ISO 15216-1
MS2	ATTCCGACTGCGAGCTTATT	ATTCCCTCAGCAATCGCAGCAAACT	TTCGACATGGGTAATCCTCA	17
*E. coli* ybbW	TGATTGGCAAAATCTGGCCG		GAAATCGCCCAAATCGCCAT	18

**Table 2 viruses-14-00762-t002:** A single cluster of GII HuNoV was detected in the Sound between January and April 2020 using the Bernouilli model available in SaTScan v9.6.1.

Cluster	Sites	RT-PCR Confirm	Cluster Size (km)	Time Frame	OvE	*p*-Value
A	NV1	GII.2	0	8 April to 23 April 2020	10.44	0.006

## Data Availability

Not applicable.

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
