# Peer review of "Spatial and Temporal Pattern of Norovirus Dispersal in an Oyster Growing Region in the Northeast Pacific"

_viruses, 2022, doi:10.3390/v14040762_

Round 1

Reviewer 1 Report

The paper by T. Green et al. investigates the spatiotemporal dispersion of human norovirus (HuNoV) in a coastal region on the Western coast of Canada through deployment and testing of sentinel oysters. It reports a low prevalence of HuNoV in tested oysters and a unique cluster of contamination  with a dispersion radius below 7km.

Overall the paper is well written and organized, the study design is adapted, conclusions clearly presented. However, some improvements are needed to better understand the results.  

The study reports a high amount of work with more than 500 pools of oysters analyzed for HuNoV GI and GII, representing 12 sites investigated 6 times at 2-weeks intervals. All this is summarized in a single value of global HuNoV prevalence in oysters, without intermediate results. Tables listing the number of positive / inconclusive / negative pools for each site and 2-weeks periods for NoV GII and E. coli, are needed to allow critical reading of the paper. Especially since the high number of inconclusive results may lead to underestimating HuNoV prevalence, as discussed by the authors.

L37. There are now ten confirmed NoV genogroups, see Chhabra et al, 2019.

L52-55 : this focus on Canada is understandable given the study design, but figures on shellfish-borne NoV burden in other countries / worldwide would allow the reader to better understand the global context.

L.141 : “the ΔCq < 3 between neat and five-fold diluted cDNA sample” : is this how PCR inhibition was considered ? If so what is the link between the dilution factor and the ΔCq threshold ? Usually, inhibition is calculated by comparing the qPCR efficiency (using the slope of the standard curve in Cq/log10(genome copies) and the ΔCq between neat and 10-fold diluted sample. Besides, if the ΔCq is low, like 1 or even 0, it means that the sample is inhibited, so instead of keeping samples with Cq below a threshold, one would expect to keep ΔCq higher than a threshold (here > log2(5) ie, 2.3). Please explain whether inhibited samples were excluded or considered as inconclusive, in the case of a positive qPCR result or a negative one.

L.183 Why was the Bernouilli model selected ? Has it been applied to similar datasets/questions previously ? How does the model take into account inconclusive results ?

L.290. Is epidemiological information on acute gastroenteritis prevalence, at the time of the study and in the study area, available to inform more precisely on this potential impact ? Also using your dataset, does the prevalence of positive oyster pools decrease from mid-March 2020 onwards ? In France, a study reported HuNoV contamination of oysters at the same period and afterwards, despite pandemic counter-measures (Desdouits et al, 2021). Given the high persistence of NoV in shellfish, one month of these measures may not have impacted the results so much… but it is indeed a factor to consider and discuss.

Author Response

Reviewer 1:

The study reports a high amount of work with more than 500 pools of oysters analyzed for HuNoV GI and GII, representing 12 sites investigated 6 times at 2-weeks intervals. All this is summarized in a single value of global HuNoV prevalence in oysters, without intermediate results. Tables listing the number of positive / inconclusive / negative pools for each site and 2-weeks periods for NoV GII and E. coli, are needed to allow critical reading of the paper. Especially since the high number of inconclusive results may lead to underestimating HuNoV prevalence, as discussed by the authors.

To assist the readers, we now provide a new figure that presents the estimated prevalence of GII HuNoV using both positive and positive + inconclusive cases for the calculation of prevalence. We use a figure as suggested by reviewer 2.

Updated Text: Figure 1 shows the estimated prevalence for GII HuNoV in the Sound using both positive and inconclusive cases.

L37. There are now ten confirmed NoV genogroups, see Chhabra et al, 2019.

Line 37 has been updated as per the reviewer’s suggestion.

Updated Text: There are at least 49 norovirus genotypes classified within ten genogroups (GI to GX) [3].

L52-55 : this focus on Canada is understandable given the study design, but figures on shellfish-borne NoV burden in other countries / worldwide would allow the reader to better understand the global context.

Line 52-55 has been updated as per the reviewer’s suggestion.

Updated text: Although the focus of this study is on Canada, HuNoV contamination of oyster is a global problem with the United Kingdom reporting ~11,800 cases of HuNoV per year linked to consumption of contaminated oysters [13].

L.141 : “the ΔCq < 3 between neat and five-fold diluted cDNA sample” : is this how PCR inhibition was considered ? If so what is the link between the dilution factor and the ΔCq threshold ? Usually, inhibition is calculated by comparing the qPCR efficiency (using the slope of the standard curve in Cq/log10(genome copies) and the ΔCq between neat and 10-fold diluted sample. Besides, if the ΔCq is low, like 1 or even 0, it means that the sample is inhibited, so instead of keeping samples with Cq below a threshold, one would expect to keep ΔCq higher than a threshold (here > log2(5) ie, 2.3). Please explain whether inhibited samples were excluded or considered as inconclusive, in the case of a positive qPCR result or a negative one.

We have recalculated PCR inhibition in each sample using the slope of the curve between neat and 1/5 dilution for MS2 [E % = (10(-1/slope)) x 100]. Samples are considered PCR inhibited if the amplification efficiency is less than 85 %. PCR inhibited samples were re-extracted and tested again. If re-extracted samples failed a second time, the sample was excluded from statistical analysis. Changes to the text are listed below.

Updated text: Samples were considered valid when the extraction efficiency of MS2 was greater than 5 % and the amplification efficiency of MS2 was >85%, calculated from the slope of the curve between neat and five-fold diluted cDNA sample (Efficiency = 1+1. Samples that did not meet the extraction- or amplification-efficiency for MS2 were re-extracted and tested again from oyster digestive tissue stored at -80oC. Re-extracted samples that did not meet the criteria for extraction- or amplification-efficiency were not considered for statistical analysis.

L.183 Why was the Bernouilli model selected ? Has it been applied to similar datasets/questions previously ? How does the model take into account inconclusive results ?

The SatScan statistical model is gaining popularity in aquatic animal health studies for evaluating spatial and temporal dispersal of pathogens in coastal waters (examples are Godoy et al., 2013 Virol. J. 10:344-; Cantrell et al., 2021 Aquaculture Env. Inter. 13:65-; Vanderstichal et al., 2015 Aquaculture 437:120-). The SatScan user manual recommends to use the Bernoulli model when there is case and non-case data for space-time scan statistics. The references above also use the Bernoulli model.

L.290. Is epidemiological information on acute gastroenteritis prevalence, at the time of the study and in the study area, available to inform more precisely on this potential impact ? Also using your dataset, does the prevalence of positive oyster pools decrease from mid-March 2020 onwards ? In France, a study reported HuNoV contamination of oysters at the same period and afterwards, despite pandemic counter-measures (Desdouits et al, 2021). Given the high persistence of NoV in shellfish, one month of these measures may not have impacted the results so much… but it is indeed a factor to consider and discuss.

Unfortunately, we do not have epidemiological data for the prevalence of gastroenteritis during the study period. The original goal was to collect wastewater samples during this period of time, but there are large gaps in our sampling as public health officials scrambled to deal with SARS-COV-2. We do not see a decrease in the prevalence of HuNoV. We have updated the results to highlight that the estimated prevalence was highest at the end of the trial (+ Figure 1). We have also updated the text to include the observations from France.

Updated text: The estimated prevalence for GII HuNoV in the Sound changed throughout time, with the fifth sample-period (0.1 ± 0.8%) having the lowest prevalence and the sixth sample period (6.8 ± 1.7%) having the highest prevalence.

We did detect GII.2 HuNoV within the Sound on the 23rd of April, 2020 and this sample period coincided with the highest prevalence of GII HuNoV in the Sound (Figure 1). During the first wave of SARS-COV-2 (May 2020), French oysters from the Atlantic and Mediterranean coasts were also contaminated with HuNoV [35].

Reviewer 2 Report

Manuscript ID: viruses-1612442

Thank you for your interesting manuscript. However, the manuscript lacks a lot of data to prove the study. Therefore, there are some major comments which need to revise and add to the manuscript to improve the quality of the study as follows

Major comments:

  1. The manuscript is lack of results and Figures which make the manuscript harder for understanding. The authors should check and add some necessary images.
  2. There are not shown the RT-PCR results, standard curve, LOD of detection which is very important to data. They should be doing extra experiments and adding to the manuscript.
  3. What is the specificity of the RT-PCR?
  4. Besides, the overall data of this manuscript is really weak. The authors strongly comment to add more data to improve the quality of the manuscript to be suitable for publication on Viruses

Minor comments:

  1. The keywords are not the format style and also “keyword 1”. Please check and correct them.
  2. There are a lot of typos (e.g. temperature symbol (lines 108, 122, 136); reference styles (ref. 30)

Author Response

Reviewer 2

Major comments:

  1. The manuscript is lack of results and Figures which make the manuscript harder for understanding. The authors should check and add some necessary images.

We have provided an additional figure to help the reader interpret the data. Figure 1.

  1. There are not shown the RT-PCR results, standard curve, LOD of detection which is very important to data. They should be doing extra experiments and adding to the manuscript.

We have provided additional information in the manuscript to highlight the dynamic range of our standard curves were 5x105 to 5x101copies of the target gene, and we typically had an amplification efficiency of 100% and R2 >0.999. The standard curves we produced were comparable to commercially purchased curves from PRIMER DESIGN (GENESIG Norovirus Real-time detection kit for norovirus – Z-Path-Norovirus-STD). The overwhelming majority of our positive samples had a Cq value below our limit of quantification.

Updated text: The dynamic range of the standard curves were 5 x 105 to 5 x 101 copies.

  1. What is the specificity of the RT-PCR?

Oysters were tested for GI and GII HuNoV according to the ISO 15216-1_2017-03 method “microbiology of food and animal feed – horizontal method for determination of HAV and NoV in food using real-time PCR” with minor modification. We used the primers outlined in this protocol, which have been validated.

Further details about the validation of this method is presented in

Lowther, J.A. et al., 2019. Validation of EN ISO method 15216-part 1 – quantification of hepatitis A virus and norovirus in food matrices. Int. J. Food Microbiol. 288:82-

  1. Besides, the overall data of this manuscript is really weak. The authors strongly comment to add more data to improve the quality of the manuscript to be suitable for publication on Viruses

As per reviewer 1 and 2 suggestions, we have now provided a new figure 1 that provides the estimated prevalence of GII HuNoV at each sample location and time-point. We have also provided additional text. Please see comments to reviewer 1 to see a list of changes.

Minor comments:

  1. The keywords are not the format style and also “keyword 1”. Please check and correct them.

Thank you. We have made the edits.

  1. There are a lot of typos (e.g. temperature symbol (lines 108, 122, 136); reference styles (ref. 30)

Thank you for raising the temperature symbols. We believe this is an issue with the font set by the journal. The editor can please advise. We have corrected reference 30.

Round 2

Reviewer 1 Report

Thanks to the new Fig.1, the data are better presented. The representation of “inconclusive” results is especially important in Fig1B, as these lead to a very different picture for NoV GII frequency, that was not accessible in the first version of the paper.

In the abstract, it is stated that the method ISO 15216-1:2017 was followed with minor modifications. Yet in this method, divergent qRT-PCR results in replicates should not be considered inconclusive, but used separately to calculate a mean NoV concentration. Why were the results interpreted differently here? Given the very high Ct values, discrepancies in qRT-PCR results are expected and often encountered when analysis NoV in shellfish. The authors should consider positive all samples where at least one Ct was above threshold, calculate the corresponding frequencies and re-run the clustering analysis accordingly. Indeed, it is possible that some clusters were overlooked as they did not appear with the initial restrictive definition of positive samples.

If not possible, the authors should at least explain and justify this important difference between the ISO method and their own interpretation of the results in the material and methods section, and discuss the impact of this choice on the main result, i.e. estimating the distance of NoV dispersal in coastal settings.

The authors answered all other remark adequately.  

Author Response

The goal of our study was to identify potential points sources for NoV contamination within our study area (Baynes Sound, Canada) and determine how far NoV can disperse. Including the “inconclusive” samples in the analysis results in a higher background, reducing the ability to detect significant clusters (observed versus expected). We have performed the spatiotemporal analysis using positive + inconclusive samples. We still only detect one positive cluster at sample site 1 (NV1), but the p-value increases to 0.032. We would have the same conclusions that GII NoV disperses less than 7 km.

Table 1: Positive Samples

Cluster

Sites

RT-PCR Confirm.

Cluster Size (km)

Time Frame

OvE

p-value

A

NV1

GII.2

0

Apr 8 to Apr 23, 2020

10.44

0.006

Table 2: Positive + Inconclusive Samples

Cluster

Sites

RT-PCR Confirm.

Cluster Size (km)

Time Frame

OvE

p-value

A

NV1

GII.2

0

Apr 8 to Apr 23, 2020

4.59

0.032

Changes to the text are listed below.

Abstract

Line 20 – delete “minor”

Results

Line 236-238 - The spatiotemporal analysis was also repeated with the inconclusive samples as positive cases in the model. Again, only one significant cluster of GII HuNoV was identified at NV1 between April 8th to April 23rd, 2020.

Reviewer 2 Report

All the comments have been revised.

Author Response

No changes requested.